# Circulating Serum VEGF, IGF-1 and MMP-9 and Expression of Their Genes as Potential Prognostic Markers of Recovery in Post-Stroke Rehabilitation—A Prospective Observational Study

**DOI:** 10.3390/brainsci13060846

**Published:** 2023-05-23

**Authors:** Lidia Włodarczyk, Natalia Cichoń, Michał Seweryn Karbownik, Luciano Saso, Joanna Saluk, Elżbieta Miller

**Affiliations:** 1Department of Neurological Rehabilitation, Medical University of Lodz, Milionowa 14, 93-113 Lodz, Poland; 2Biohazard Prevention Centre, Faculty of Biology and Environmental Protection, University of Lodz, Pomorska 141/143, 90-236 Lodz, Poland; natalia.cichon@biol.uni.lodz.pl; 3Department of Pharmacology and Toxicology, Medical University of Lodz, Żeligowskiego 7/9, 90-752 Lodz, Poland; michal.karbownik@umed.lodz.pl; 4Department of Physiology and Pharmacology “Vittorio Erspamer”, Sapienza University, P. le Aldo Moro 5, 00185 Rome, Italy; 5Department of General Biochemistry, Faculty of Biology and Environmental Protection, University of Lodz, Pomorska 141/143, 90-236 Lodz, Poland; joanna.saluk@biol.uni.lodz.pl

**Keywords:** stroke, recovery, biomarker, rehabilitation

## Abstract

The key period in post-stroke recovery is the first three months due to the high activity of spontaneous and therapeutic-induced processes related to neuroplasticity, angiogenesis and reperfusion. Therefore, the present study examines the expression of VEGF, IGF-1 and MMP-9 proteins and their genes to identify biomarkers that can prognose brain repair ability and thus estimate the outcome of stroke. It also identifies possible associations with clinical scales, including cognitive assessment and depression scales. The study group comprised 32 patients with moderate ischemic stroke severity, three to four weeks after incident. The results obtained after three-week hospitalization indicate a statistically significant change in clinical parameter estimations, as well as in MMP9 and VEGF protein and mRNA expression, over the rehabilitation process. Our findings indicate that combined MMP9 protein and mRNA expression might be a useful biomarker for cognitive improvement in post-stroke patients, demonstrating 87% sensitivity and 71% specificity (*p* < 0.0001).

## 1. Introduction

Stroke still remains a principal cause of adult disability and mortality, and it is regarded as a priority for both the World Health Organization (WHO) and the United Nations to decrease the burden of non-communicable diseases [1,2]. The traditional definition of stroke was mainly clinical, comprising a sudden loss of focal neurological function due to infarction or hemorrhage lasting longer than 24 h; however, the updated definition integrates clinical and tissue criteria and is based on neuropathological, neuroimaging and/or clinical indications of permanent injury of brain, spinal cord or retinal cell [3]. Despite the use of increasingly effective early treatment strategies, such as intravenous tissue plasminogen activator (tPA) or endovascular thrombectomy (EVT) [4], post-stroke recovery is often incomplete and recovery rates of neurological function differ. 

Stroke rehabilitation depends on both spontaneous and therapeutic-induced processes of recovery, with important roles played by, inter alia, the extent of a stroke, regeneration, angiogenesis, neuroplasticity, pharmacotherapy, and neurorehabilitation. The first three months are the most important period in post-stroke rehabilitation, during which time spontaneous and therapeutic-induced processes of recovery are most effective. From a clinical point of view, it is important to know which patient may demonstrate the greatest potential improvement in both cognitive and functional performance several months later.

Therefore, there is a need to identify biomarkers that can be used as prognostic tools to estimate the brain repair ability in a particular patient and also as predictive tools for monitoring the response to individually tailored rehabilitation interventions [5]. Consequently, the findings may support the development of new algorithms to estimate the outcome of stroke patients and subsequent treatment. For example, upon admission to hospital, patients could be tested against a panel of biochemical and genetic laboratory tests, which are aimed at genes and proteins involved in recovery processes such as neuroplasticity [6]. 

Post-stroke recovery mechanisms remain poorly understood; however, circulating molecules related to neuroplasticity, such as MMP-9, VEGF and IGF-1, can influence the outcome when accompanied by extensive rehabilitation [6]. Molecular pathways involved in brain repair after stroke are supposed to vary over time and those related to neuroplasticity become relevant later in the subacute phase [7]. Furthermore, we hypothesize that serum molecular differences between patients might provide a possible neurophysiological and biochemical basis for differences in treatment responsiveness. Studies on animal models indicate that changes in circulating proteins and expression of their genes correlate with post-stroke recovery, and this has been supported to some degree by clinical research studies [8,9,10,11,12]. 

It is generally believed that stroke recovery may be associated with the genes and proteins known to play significant roles in neuroplasticity. For example, matrix metalloproteinase 9 (MMP-9), a zinc-dependent endopeptidase, is associated with both physiological and pathological tissue reorganization or remodeling, including neovascularization. Its levels are upregulated in post-stroke patients, which correlates with a poorer outcome [13,14,15]. It has been shown that in post-stroke patients the concentration of MMP9 is significantly higher compared to healthy volunteers [16,17]. Furthermore, MMP-9 levels appear to have predictive value for hemorrhagic transformation, with the highest level of MMP-9 recorded six hours after stroke onset, and an increased level of MMP-9 being associated with infarct exacerbation and hemorrhage conversion [18]. Furthermore, in cases of acute stroke, the serum MMP-9 level was found to predict post-stroke cognitive impairment three months later [19].

Vascular endothelial growth factor (VEGF) is a well-known strong angiogenic factor that was first described as specific for endothelial cell mitogen [20]. VEGF has a well-documented role in the nervous system, where it promotes neuroplasticity, including nerve repair, neurogenesis, and glial growth [21]. During ischemic stroke, VEGF gene expression is upregulated as a result of hypoxia; the protein usually binds to the VEGF second receptor, which is a crucial mediator of angiogenesis and increased permeability. The second receptor may stimulate the production of matrix metalloproteinases (MMPs) and induce the release of endothelial growth factors [22]. Increases in circulating VEGF have been shown to aid neurological recovery in animal models of acute stroke [9,10]. An animal model study showed that treadmill exercise may promote post-stroke recovery by regulating the caveolin-1/VEGF pathway in the ischemic zone [23]. In addition, higher exercise intensity has been found to be related to increased VEGF levels in post-stroke rehabilitation [24]. VEGF expression was found to be significantly elevated for the 90-day period after stroke onset, and this may have a significant influence on the functional outcome for cardioembolic infarction origin stroke with higher VEGF values predicting poorer outcomes [25]. 

The insulin-like growth factor 1 (IGF-1) protein is structurally and functionally similar to insulin and as a neurotrophic hormone, it plays a crucial role in the development and maturation of the central nervous system. IGF-1 serum level has been found to be influenced by stroke, with a decrease in IGF-1 in acute stroke being related to a positive outcome [26]. However, IGF-1 only has a modest influence on long-term post-stroke recovery persisting after seven years, as reflected in the mRS score distributions at all time points and is not associated with mortality or recurrent stroke [27]. It has also been confirmed that IGF-1 serum level plays a key role in short-term outcomes (three months), whereas in the longer term, e.g., two years, the associations are weakened and attenuated by other factors, such as diabetes, smoking, hypertension and hyperlipidemia [28].

Hence, the aim of the present study was to evaluate the prognostic value of proteins VEGF, IGF-1 and MMP-9 and the expression of their genes as markers of recovery in stroke patients. The expression data was compared with patient demographics and various clinical scales, including cognitive assessment and depression scales, with the goal of identifying a recovery biomarker associated with individual functional domains. The study was performed in the early subacute phase of stroke, i.e., from seven days to three months, according to the Consensus Statements from the Stroke Recovery and Rehabilitation Roundtable [7]. 

## 2. Materials and Methods

### 2.1. Subject Presentation 

A series of patients was prospectively collected for the study Figure 1; all had been admitted to the Department of Neurological Rehabilitation Medical University of Lodz, Poland, from January 2022 to January 2023. 

The minimum number of patients needed for enrollment was 28, based on sample size estimation with the use of G*Power 3.1.9.2 software [29]. The one-way repeated-measure analysis of variance (ANOVA) was used to demonstrate the medium effect size (Cohen’s *f* = 0.25) with a statistical power of 0.8, statistical significance of 0.05, and correlation between repeated measures of 0.6 (as indicated in preliminary results for improvement in physical functioning and changes in MMP-9 levels).

A total of 32 patients were enrolled, all of whom with moderate severity ischemic stroke, three to four weeks after the incident. The mean age was 68.3 (±9.1), the other clinical and demographic characteristics of the patients are shown in Table 1. 

The diagnosis of cerebrovascular event was based on clinical and neuroimaging criteria according to the updated definition of stroke [3]. Exclusion criteria included the following: intracerebral hemorrhage, chronic or significant acute inflammatory factors, neurological illness other than ischemic stroke, severe cardio-vascular disease such as myocardial infarction within less than 30 days or unstable angina pectoris, decompensated metabolic or endocrine diseases, or severe general condition with respiratory failure. The patients underwent a rehabilitation program provided by a physiotherapist, every day for a period of three weeks with weekend gaps. The post-stroke neurorehabilitation program consisted of a 60 min neurophysiological session in the morning (30 min of shaping techniques—based on daily living activities and 30 min of repetitive task practice or balance), 15 min of psychotherapy and 30 min aerobic training (two or three times a day for 10 min at 60 min intervals). However, training time was individually modified depending on physical condition and improvement in function, if necessary. 

### 2.2. Clinical Parameter Determination

#### 2.2.1. Cognitive Assessment

Cognitive assessment was performed using the Mini Mental State Examination (MMSE) and Montreal Cognitive Assessment (MoCA), depending on the patient’s age. The MMSE or Folstein test is a 30-point questionnaire most widely used in screening for cognitive impairment. It requires vocal responses, following verbal and written commands, writing a sentence spontaneously and coping with a complex figure, and provides information about orientation, memory, attention, learning, calculation, delayed recall, and construction [30]. The MMSE is regarded to have acceptable validity as a screening instrument for vascular dementia in post-stroke patients with a sensitivity of 0.96 and specificity of 0.83, assuming a cut-off score of 23/24 [31]. The MoCA is a one-page, 30-point test which evaluates the following domains: short-term memory, visuospatial abilities, executive functions, attention and working memory, language, and orientation to time and space. The threshold for normal cognitive function is ≥26. The tool is widely used as a brief cognitive tool in both the acute/subacute and chronic post-stroke terms [32], and a MoCA score of ≤23 allows prediction of post-stroke cognitive impairment and functional dependence after stroke [33,34].

#### 2.2.2. Depressive Symptom Assessment 

Depressive symptoms were assessed using The Geriatric Depression Scale (GDS) and Beck’s Depression Inventory (BDI), depending on the patient’s age. The GDS is a practical, self-reporting screening tool used in clinical settings to evaluate depression in older adults. The present study used the GDS short version; this version has adequate reliability and validity compared to the long version [35]. It consists of 15 “Yes/No” questions. Final scores of 0–4 are considered normal (depending on age, education and complaints), and scores of 5 and above indicate depression: 5–8 being mild, 9–11 being moderate and 12–15 indicating severe depressive symptoms [36]. 

The BDI is a widely used 21-question self-reporting inventory measuring the severity of depression in adolescents and adults. Each question has a set of at least four possible answer choices reflecting the intensity of symptoms with a value of 0 to 3 assigned for each answer. The total score determines the severity of depression as follows: 0–9 indicates minimal depression, 10–18 indicates mild depression, 19–29 indicates moderate depression, and 30–63 indicates severe depression [37,38]. 

#### 2.2.3. Physical and Motor Condition Assessment 

The National Institutes of Health Stroke Scale (NIHSS) is recommended as a valid tool to assess stroke severity; it is also considered to be a strong predictor of outcomes after stroke [39]. The NIHSS includes the following domains: level of consciousness, eye movements, integrity of visual fields, facial movements, arm and leg muscle strength, sensation, coordination, language, speech and neglect. Each domain is scored from 0 to 2, 0 to 3, or 0 to 4. The total score ranges from 0 to 42, and a higher score is associated with more severe stroke [40]. 

The Barthel Index (BI) is a measure of independence in activities of daily living (ADL) The 10 items included in the BI pay attention to self-care and mobility. Total possible scores range from 0–20, with lower scores indicating increased disability, and a score of 17 and above representing normal activity [41].

The Modified Rankin Scale is a global outcome rating scale that measures independence rather than the result of specific tasks; it also reflects both mental and physical adaptations to neurological deficit. The scale consists of six grades (from 0 to 5), with 0 indicating no symptoms and 5 corresponding to severe disability [42].

#### 2.2.4. Blood Sample Collection 

Whole blood samples were collected between 7 am and 9 am, under dietary fasting conditions. Study material was taken twice: before therapy and in the fourth week of hospitalization in the study group, and once prior to treatment in control. The research material was placed into CPDA1 containing tubes (Sarstedt, Nümbrecht, Germany). Immediately after collection, part of the sample was frozen at −80 °C to measure the mRNA expression. Plated pour plasma was obtained by differential centrifugation (15 min, 1500× *g*, 25 °C). All samples were collected and stored using protocol.

### 2.3. Determination of IGF1, VEGF and MMP-9 Level in Plasma

Measurement of MMP-9, VEGF, and IGF1 protein content was conducted using a Human IGF1 SimpleStep ELISA^®^ Kit (Abcam, Cambridge, UK), Human VEGF SimpleStep ELISA^®^ Kit (Abcam, Cambridge, UK), and Human MMP-9 ELISA Kit (Invitrogen, Waltham, MA, USA), respectively. All procedures were conducted in accordance with the manufacturer’s protocol. The endpoint reading (color intensity) of all analytes was measured at 450 nm and protein concentration was calculated from the standard curve.

### 2.4. Determination of MMP-9, VEGF-A and IGF1 Expression in Whole Blood Samples

RNA isolation and purification was performed using a DNA/RNA Extracol Kit (EURx, Gdansk, Poland). RNA purity and quantity was first estimated using a Synergy HTX Multi-Mode Microplate Reader, equipped with a Take3 Micro-Volume Plate (BioTek Instruments, Inc., Winooski, VT, USA). Following this, the diluted RNA samples were transcribed into cDNA using a High-Capacity cDNA Reverse Transcription Kit (Applied Biosystems, Waltham, MA). All steps were performed in accordance with the manufacturer’s protocol. Gene expression was determined by RealTime PCR using a TaqMan Universal Master Mix II (Life Technologies, Carlsbad, CA, USA) in a CFX96 real-time PCR system (BioRad Laboratories, Hercules, CA, USA). Expression levels were obtained using TaqMan probes (Hs01547656_m1 for human IGF-1, Hs00900055_m1 for human VEGF-A, Hs00957562_m1 for human MMP-9, and for endogenous control (GAPDH)—Hs99999905_m1) (Life Technologies, Carlsbad, CA, USA). All procedures followed the manufacturer’s protocol. 

### 2.5. Data Analysis 

The dataset presented 210 (12.6%) missing data points. These were imputed with the use of multiple imputation by chained equation procedure under the *missing at random* assumption before the data analyses were performed. Descriptive statistics for the continuous and ordinal variables included arithmetic mean with standard deviation (for the sample characterization) or standard error of mean (for the purpose of estimation), whereas dichotomous variables were expressed as numbers and frequencies. mRNA gene expression was presented as –ΔCt to make the distribution closer to normal. One-way repeated-measure ANOVA was used to compare post- and pre-rehabilitation outcomes. Additional covariates were optionally added linearly linked with the outcome in the ANOVA models. The other general linear models were used for selected purposes. As distribution normality was violated in a few instances, respective non-parametric tests were also performed; these are reported in Appendix A. Multivariate logistic regression modeling was performed to estimate the predictive values of the selected pre-rehabilitation biochemical parameters in forecasting post-rehabilitation psychological outcomes. Internal validation of the proposed models was performed using the 10-fold cross-validation technique. The modeling results were additionally presented with the use of receiver operating characteristic (ROC) curves. *p*-values below 0.05 were considered statistically significant. The analysis was performed using STATISTICA 13.3 Software (StatSoft; Tulsa, OK, USA). 

## 3. Results

After the three-week rehabilitation program, both the physical and psychological condition of the patients improved significantly as compared to pre-rehabilitation time. Details are presented in Table 2.

Among the tested proteins and mRNAs, significant changes in plasma MMP-9 and VEGF protein content were observed over the rehabilitation process. MMP-9 protein concentrations fell from 112.2 ± 9.7 pg/mL before rehabilitation to 85.4 ± 10.6 pg/mL afterwards, i.e., by 24%, whereas VEGF protein level increased from 44.8 ± 25.9 pg/mL to 66.3 ± 50.3 pg/mL, i.e., by 48% (Figure 2). 

The reported changes in MMP-9 and VEGF protein plasma concentration observed over the rehabilitation process were correlated with improvements in cognitive function and amelioration of depressive symptoms. However, although MMP-9 decreased during rehabilitation, its rise predicted better cognitive outcome and although VEGF increased during the rehabilitation, its fall was associated with less depressive symptoms. Moreover, although a negligible change in VEGF-A mRNA expression was observed over the rehabilitation process, it significantly correlated with an amelioration of depressive symptoms (Table 3). 

To gain a predictive insight into the improvement of psychological function over the rehabilitation process, the pre-rehabilitation parameters were compared with psychological improvement indices. Pre-rehabilitation mRNA expression of MMP-9 was found to be inversely linked to cognitive improvement, whereas baseline VEGF protein level predicted improvement in depression symptoms. 

The MMP-9 and VEGF protein levels fluctuated during the rehabilitation process, and these proteins together with respective mRNAs could predict some improvements in cognitive function and depressive symptoms over the rehabilitation process, respectively (Table 3 and Table 4). Therefore, pre-rehabilitation MMP-9 protein and mRNA levels were used to model the odds for cognitive improvement after rehabilitation, whereas VEGF protein and mRNA levels were used to model the chance of amelioration of depressive symptoms. Both MMP-9 and VEGF were suitable for inclusion into the model at both the mRNA and protein levels, as the correlations between respective mRNA and protein were negligible: r = 0.20, *p* = 0.2661 for MMP-9 and r = −0.00, *p* = 0.9794 for VEGF. 

A multivariate logistic regression analysis with internal validation was applied. An improvement in the psychometric indices by at least three points was the modeled outcome. Both models were adjusted with the additional inclusion of basic sociodemographic parameters (age and sex) as covariates to increase patient-specific relevance. It was found that pre-rehabilitation MMP-9 protein and mRNA levels could predict post-rehabilitation cognitive improvement. The model fitted well to the data and was not overfitted; sensitivity and specificity measures were promising, even in the validation set. In addition, the Hosmer–Lemeshow test results were insignificant, indicating that it could work well in all population subgroups (Table 5 and Figure 3A). 

On the other hand, in multivariate logistic regression analysis, VEGF protein and mRNA levels failed to predict any amelioration in depressive symptoms, as indicated by insignificant predictors and unsatisfactory model indices (Table 6 and Figure 3B). 

## 4. Discussion

One of the most difficult clinical challenges encountered during post-stroke rehabilitation is the unpredictable, multisymptomatic course of the condition. However, the results of treatment in various medical areas can be better predicted by including biomarkers. Our present findings suggest that the combined protein and mRNA expression of MMP-9 may be a useful biomarker for cognitive improvement in post-stroke patients, assuming an increase of three points on the MMSE scale, and that this test offers 87% sensitivity and 71% specificity (Figure 3A).

Post-stroke cognitive impairment and dementia (PSCID) is one of the most important problems influencing functional recovery and quality of life and can be a major source of morbidity and mortality after stroke [43]. Although a number of biomarkers related to neuroimaging are ready to be used in clinical trials, they are envisaged for implementation in motor and language functional domains; there are still no neuroimaging biomarkers intended for somatosensory problems and cognition [44]. Hence, there is a need for biomarkers of cognitive recovery in the biochemical and molecular fields.

In the classic sense, as defined by Naylor, a biomarker should be impartially measured and evaluated as a marker of both physiological and pathological processes, and also as an indicator of response to a therapeutic intervention [45]. Furthermore, a potentially useful biomarker must fulfil several important criteria, including sensitivity, specificity, repeatability of measurements, safety of collection, cost-effectiveness and quickness [46].

Our findings indicate that MMP-9 has predictive potential as a useful biomarker of stroke recovery in the field of cognition. Although MMP-9 involved in the degradation of the extracellular matrixes is considered to cause cerebral damage after stroke, studies confirm that MMP-9 also plays a role in synaptic plasticity and synapse remodeling [47]. In humans, the MMP family consists of 23 members, including secreted and transmembrane proteins. In the central nervous system, MMPs are synthesized and secreted by neurons and glia in an inactive pro-enzyme form (zymogen) and are later converted into active enzymes [48]. The levels of both the pro-form and active form of MMP-9 were found to increase after the induction of late-phase long-term potentiation (LTP), i.e., persistent strengthening of synapses in the CA1 area of the hippocampus, which is the physiological basis of synaptic plasticity [49]. Studies also indicate that MMP-9 is a crucial dendritic spine shape modulator [50]. Importantly, MMP-9 has been found to have a regulatory role in synaptogenesis, myelination and axonal outgrowth [51].

MMP-9 also plays a key role in the transformation of pro-BDNF into the mature form of BDNF in the hippocampus [52].

Interestingly, a study examined hypoxia response element (HRE)-regulated MMP-9 and its effect on glial scars and neurogenesis in delayed ischemic stroke, seven days after transient middle cerebral artery occlusion (tMCAO) based on a rat model. HRE-MMP-9 improved neurological outcomes at three and five weeks after tMCAO, reduced ischaemia-induced brain atrophy, and degraded glial scars (*p* < 0.05). Furthermore, HRE-MMP-9 increased the number of microvessels in the peri-infarct area (*p* < 0.001) [8]. These findings confirm that MMP-9 has a beneficial effect in the subacute phase of stroke, suggesting that its influence may be related to neuroplasticity and angiogenesis. This statement is consistent with our present findings suggesting that the combined protein and mRNA expression of MMP-9 may be a useful biomarker for cognitive improvement in the subacute phase.

Conversely, elevated MMP-9 serum concentrations compared to controls have been noted in the hyperacute period of stroke; however, MMP-9 also appears to play a role in damaging the blood-brain barrier and regulating inflammation [53,54]. This, in turn, is related to poor outcome [14,15]. Furthermore, those pathological processes probably explain recent results indicating that MMP-9 levels measured in the acute phase of stroke have predictive value for post-stroke cognitive impairment (PSCI) three months after stroke [19]. Furthermore, studies suggest that at high levels, there are reports that pro-inflammatory and atherosclerosis signaling factors, including MMP-9, were associated with a significantly higher risk of adverse clinical outcomes.

Guo et al. propose that MMP-9, S100A8/A9, high sensitivity C-reactive protein (hsCRP) and growth differentiation factor-15 (GDF-15) are effective predictors of prognosis at three months after ischemic stroke and may be useful for risk stratification among ischemic stroke patients [55]. Our findings indicate that MMP-9 serum levels decreased by 24% (*p* = 0.032) in stroke patients after a three-week course of rehabilitation, both as raw data and following adjustment for sex and age. The reported changes in MMP-9 occurring during rehabilitation were found to be correlated with an improvement in cognitive function; more specifically, its rise predicted better cognitive outcomes, which may indicate the presence of neuroplasticity promoted by MMP-9.

Zhong et al. demonstrated a linear association between MMP-9 levels and cognitive impairment (*p* < 0.001 for linearity). They indicate that an increased MMP-9 level in the acute phase of ischemic stroke was associated with three-month cognitive impairment, independent of adopted risk factors; this is consistent with our present data indicating that the pre-rehabilitation level of MMP-9 gene expression was negatively correlated with an improvement in cognition (Table 5) [56]. Interestingly, serum MMP-9 level has also been proposed as a significant factor for diagnosing cognitive impairment due to cerebral small vessel disease, which suggests that MMP-9 may be associated with vascular dementia [57].

Our present study indicates that pre-rehabilitation MMP-9 protein and mRNA levels could forecast a recovery in cognitive function with 87% sensitivity and 71% specificity (*p* < 0.0001) (Figure 3A), thus allowing further treatment to be planned in line with the concept of personalized medicine. By comparison, serum MMP-9 has also been found to predict hemorrhagic transformation after thrombolytic therapy in acute ischemic stroke with 92% sensitivity and 74% specificity [58]. Therefore, MMP-9 protein level appears to be a promising candidate for inclusion in future multicenter confirmatory studies as a possible marker of cognitive recovery. This conclusion was also confirmed by multivariate logistic regression analysis with internal validation, with the modeled outcome being assumed as an improvement in the psychometric indices by at least three points (Table 5).

However, any assessment of cognitive problems following stroke is complicated by the coexistence of stroke-related impairments, such as visual disturbance or paresis of the dominant hand and dysphasia; as such, it should be performed by qualified specialists. Despite these limitations, data indicates that MMSE and MoCA could be used to accurately screen for impairments at all levels of severity, and may represent the best option of screening for vascular dementia [59].

In our study, cognitive assessment was performed twice: once at the baseline and once again three weeks later, i.e., after rehabilitation MMSE and MoCA are considered to be test-retest reliable factors for monitoring cognitive function in dementia patients [60]. However, gains in neuropsychological test performance scores have been attributed to practice effects [61] and a short-term practice effect (even small) should be considered when interpreting changes in scores [62]. As reported by Feenay, in a study of 130 participants who underwent two cognitive assessments between two and four months apart, the standard error of measurement (SEM) of the MMSE was found to be one point, leading to an minimum detectable change (MDC) of three points [63]. In our study, the re-test was performed only three weeks later; therefore, the practice effect was estimated at two points for the MMSE, which was subtracted from the obtained result. 

Our findings do not confirm previous reports indicating an association between serum MMP-9 levels and post-stroke depression (Table 4). A recent study suggests that elevated serum MMP-9 levels in the acute phase of ischemic stroke were related to increased risk of post-stroke depression within three months, suggesting that MMP-9 has a prognostic role [64]. However, as blood samples were collected in the subacute phase of stroke in the present study, further research is needed in this phase of stroke. 

VEGF is another molecule involved with neuronal survival and angiogenesis. Its main role is associated with its pro-angiogenic activities, but it has also demonstrated neurotrophic and neuroprotective effects on both on the central and peripheral nervous systems [65,66]. Neurogenesis is achieved by stimulating the endothelium to release neurotrophic factors, as shown in Schwann cells after hypoxia in the peripheral nervous system [67]. It has been also demonstrated that VEGF generated by ependymal cells triggers and strengthens neuronal precursor proliferation in the central nervous system [68]. In addition, these neurogenic effects are believed to be related to an increase in the transdifferentiation of striatal astrocytes into new mature neurons [69]. Serum VEGF level increases immediately after stroke, peaks at three days post-stroke (*p* < 0.001) [70] and remains significantly higher than controls for 90 days after stroke onset [25]. In our study, a significant increase in VEGF serum level (by 48%) was observed after the three-week rehabilitation course (*p* = 0.011; Figure 2B).

VEGF level has been found to correlate with depression, and data suggests that polymorphisms in the VEGF pathway are associated with the severity of depressive symptoms and correspond with higher plasma concentrations of VEGF (*p*-value = 0.006) [71]. Interestingly, our present findings suggest that a decrease in VEGF protein plasma is correlated with an amelioration of depressive symptoms in post-stroke patients (*p* = 0.043; Table 3). Indirectly, this seems to be consistent with previous data. A series of studies indicates that increased VEGF serum level may be a strong predictor of unfavorable outcomes [72,73]. Notably, other studies have analyzed VEGF level in relation to various types of strokes. Zhang et al. [74] report that increased serum VEGF may be used as an independent predictor of a poor outcome in minor ischemic stroke. Matsuo et al. [25] divided stroke patients into four groups depending on subtype of ischemic stroke: atherothrombotic infarction (ATBI), lacunar infarction (LAC), cardioembolic infarction (CE) and other types (OT). After adjustment for possible confounding factors, VEGF plasma level was an independent predictor of unfavorable functional outcome only in CE patients. It can be concluded that many variables affect VEGF values, and further well-planned and detailed research is needed.

IGF-1 is a strong neuroendocrine regulator of CNS development, maturation and cellular neuroplasticity. Its potential mechanisms of action include modulation of glutamatergic receptor subunits, potentiation of glutamatergic transmission or alterations in calcium channel conductance [75]. It has been demonstrated that IGF-1 levels in circulation and the cerebrospinal fluid (CSF) decrease with age and a positive relationship exists between serum and CSF IGF-1 levels [76]. Furthermore, age-related reductions in IGF-1 correspond to increases in risk of hypertension, diabetes, and other cardiovascular diseases including ischemic stroke [77,78,79]. In turn, patients experiencing acute ischemic stroke demonstrate lower IGF-1 expression in comparison with healthy subjects with the amount decreasing with disease aggravation [80]. Although these literature data suggest that IGF-1 has prognostic value, our present findings indicate that IGF-1 plasma level was not affected by three-week rehabilitation following stroke (Figure 2C). IGF1 mRNA expression was only detectable in nine patients, and this parameter was excluded from further analyses.

Previous studies have also noted that changes in IGF-1 serum level are not statistically significant for assessing stroke recovery. Edinger et al. report that, based on a multivariate analysis including IGFBP-3 (insulin-like growth factor binding protein-3) IGF-1, age, thrombolysis and NIHSS, only low IGFBP-3 levels were associated with an unfavorable functional outcome (OR 7.2, 95%CI 1.8–29.0, *p* = 0.006). Additionally, after considering lesion volume, only IGFBP-3 levels were predictive of poor functional outcome (OR 7.2, 95%CI 1.5–35.5, *p* = 0.015) [81].

However, the scientific evidence regarding whether plasma levels of IGFBP-3 and IGF-1 are indicative of functional outcome remains inconsistent. Armbrust et al. report that low IGF-1 levels in the acute phase of stroke (day 8) were independently associated with a decreased risk of an unfavorable outcome [82]. Interestingly, higher levels of serum IGF-1 were observed in patients who demonstrated greater improvements in cognition after combined aerobic and cognition training, suggesting that IGF-1 may be involved in behaviorally induced plasticity [83]. In addition, higher exercise-induced levels of serum IGF-1 at the baseline were found to predict 40% of the variability in advance of fluid intelligence. It is noteworthy that only exercise-induced levels of IGF-1 were prognostic of this response and not resting levels [83]. Another study found that IGF-1 level was positively correlated with aerobic fitness in acute stroke patients: individuals with higher levels of IGF-1 demonstrated significant improvements in estimated aerobic fitness, which might be beneficial to stroke recovery [84].

The search for useful biomarkers for stroke recovery offers promise but requires further effort.

## 5. Limitations and Future Perspective

There are some limitations that should be included for data interpretation. The main limitation was the small number of stroke patients. However, it was still relatively large in comparison to previous studies analyzing a similar area of interest [14]. Moreover, another limitation was that this study included male and female participants. Our study focused on patients with moderate ischemic stroke severity; therefore, the results are characteristic to a defined level of disability, without long-term analyses.

Future research should aim to create a laboratory test panel as the use of multiple biomarkers covering several pathways has been found to improve prognostic ability. In addition, it would be worth analyzing the correlation between biomarkers and cognitive impairment not only regarding cognition as a whole, but in separation into the domains which constitute cognitive functioning. There is a clinical need for longitudinal studies of poststroke biomarkers, and very interesting research is in progress, the results of which we are waiting for [7,85,86,87,88].

## 6. Conclusions

There are no prognostic biomarkers to identify stroke recovery and to support decisions regarding stroke care and rehabilitation. In this prospective observational study, among the analyzed proteins and mRNAs, significant changes were observed in plasma MMP-9 and VEGF protein content over a 3-week follow-up time. Particularly, MMP-9 appears to be a promising molecule as a biomarker of cognitive recovery in the subacute phase, which should encourage a further multicenter confirmatory study.

## Figures and Tables

**Figure 1 brainsci-13-00846-f001:**
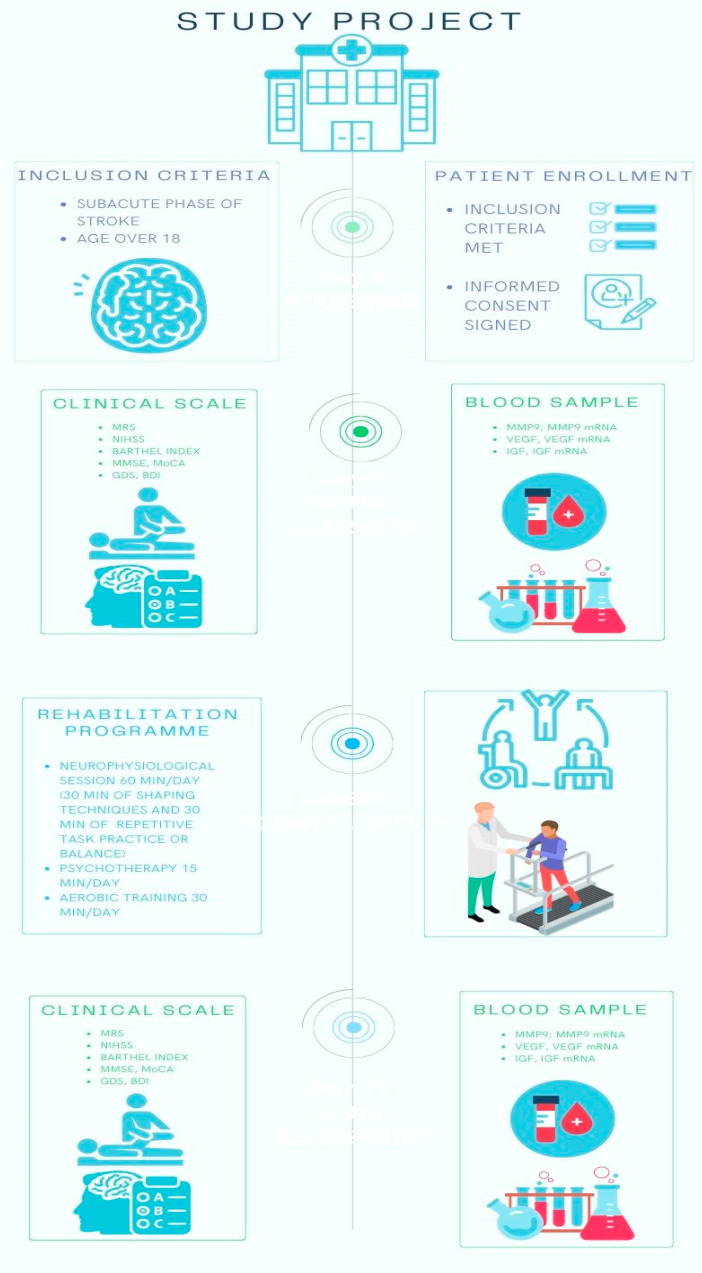
The study project. MRS—Modified Rankin Scale, NIHSS—National Institutes of Health Stroke Scale, MMSE—Mini Mental State Examination, MoCA—Montreal Cognitive Assessment, GDS—Geriatric Depression Scale, BDI- Beck Depression Inventory, MMP-9—Matrix metalloproteinase 9, VEGF—Vascular endothelial growth factor, IGF-1—Insulin-like growth factor 1.

**Figure 2 brainsci-13-00846-f002:**
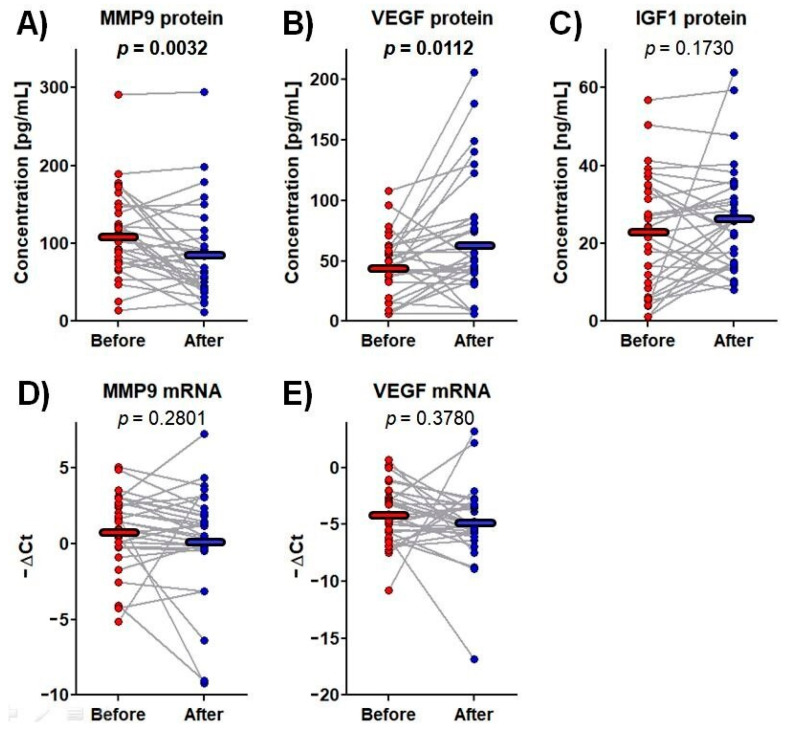
The protein concentrations and mRNA expressions of the tested genes before and after rehabilitation. Data points represent the levels of measurands for each patient, whereas the horizontal lines indicate the mean values. One-way repeated-measure analysis of variance was used. The *p*-values for significant results are presented in bold. (**A**) MMP9 protein concentration in plasma; (**B**) VEGF protein concentration in plasma; (**C**) IGF1 protein concentration in plasma; (**D**) *MMP9* mRNA expression in whole blood; (**E**) *VEGF* mRNA expression in whole blood.

**Figure 3 brainsci-13-00846-f003:**
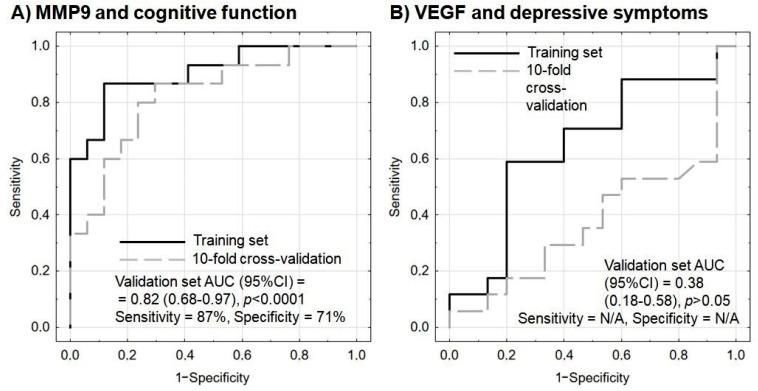
Receiver operating characteristic (ROC) curves for the performance of the model to predict (**A**) cognitive improvement based on MMP9 protein and mRNA levels, and (**B**) amelioration of depressive symptoms based on VEGF protein and mRNA levels. In each model, age and sex are included as covariates. Sensitivity and specificity are calculated based on the maximization of Youden’s index; sensitivity and specificity were not provided for the model of VEGF and depressive symptoms as the performance on the validation set was below the random guess. N/A—not applicable.

**Table 1 brainsci-13-00846-t001:** Demographic characteristics.

Parameter	Mean (SD) or Number (Frequency)
Sociodemographic	
Sex	
Female	14 (44%)
Male	18 (56%)
Age [years]	68.3 (9.1)
Comorbidity and treatment	
Hypertension	17 (53%)
Diabetes	10 (31%)
Atherosclerosis	4 (13%)
Trombolytic treatment	4 (13%)
Blood parameters	
Sodium	139.2 (3.0)
Potassium	4.2 (0.7)
WBC	7.46 (2.16)
RBC	4.30 (0.63)
Hb	12.9 (1.7)
HCT	40.2 (4.0)
Urea	29.8 (13.8)
Creatinine	0.7 (0.2)
CRP	9.0 (10.1) 6.0 (1.3–11.2) ^a^

^a^—where the distribution was highly skewed, the results are also presented as median with 1st–3rd quartiles.

**Table 2 brainsci-13-00846-t002:** Change in physical and psychological functioning before and after rehabilitation.

Parameter	Mean Estimate ± SEM		*p*-Value
	Before	After	
Physical			
ADL	8.8 ± 0.8	13.0 ± 0.9	0.0008
Rankin	3.9 ± 0.2	3.1 ± 0.2	0.0024
NIHSS	9.0 ± 0.5	6.6 ± 0.7	0.0059
Psychological			
Cognitive function			0.0004 ^a,b^
MOCA	20.0 ± 1.9	24.6 ± 1.9	
MMSE	23.4 ± 0.6	25.2 ± 0.8	
Depressive symptoms			0.0162 ^a,c^
BDI	7.6 ± 1.5	7.3 ± 2.4	
GDS	11.8 ± 0.9	6.8 ± 1.0	

^a^—as different psychometric tests were used in patients of different ages, the type of the test was used as a covariate in two-way repeated-measure analysis of variance (a between factor), ^b^—there was an insignificant within–between interaction observed (*p* = 0.1008) suggesting similar improvement in cognitive function in both age groups, ^c^—there was a significant within–between interaction observed (*p* = 0.0302) suggesting that senior participants demonstrated greater improvements in cognitive function than younger adults

**Table 3 brainsci-13-00846-t003:** Correlation between the improvement in psychological functioning over the rehabilitation process and changes in the levels of the tested protein/mRNA.

Change in the Level of Measurand over the Rehabilitation Process	Cognitive Improvement		Improvement in Depression	
	Beta Coefficient ± SEM	*p*-Value	Beta Coefficient ± SEM	*p*-Value
MMP9 protein	**0.49 ± 0.15**	**0.0034**	−0.02 ± 0.17	0.8878
VEGF protein	0.01 ± 0.18	0.9333	**−0.34 ± 0.16**	**0.0427**
IGF1 protein	−0.03 ± 0.18	0.8721	−0.07 ± 0.17	0.7083
*MMP9* mRNA	−0.02 ± 0.19	0.8974	0.29 ± 0.18	0.1076
*VEGF-A* mRNA	−0.22 ± 0.18	0.2257	**0.42 ± 0.16**	**0.0117**

Changes in the levels of measurands were calculated as the difference between post-rehabilitation and pre-rehabilitation value. Separate general linear models were built with the psychological improvement indices as outcomes, changes in measurand levels as continuous predictors, and the type of the psychometric test used as a dichotomous covariate. Significant results are presented in bold.

**Table 4 brainsci-13-00846-t004:** Correlation between the improvement in psychological functioning over the rehabilitation process and the pre-rehabilitation levels of measurands.

Pre-Rehabilitation Level of Measurand	Cognitive Improvement		Improvement in Depression	
	Beta Coefficient ± SEM	*p*-Value	Beta Coefficient ± SEM	*p*-Value
MMP9 protein	0.26 ± 0.17	0.1402	−0.16 ± 0.17	0.3504
VEGF protein	0.03 ± 0.19	0.8603	**−0.41 ± 0.17**	**0.0218**
IGF1 protein	0.08 ± 0.18	0.6575	0.11 ± 0.17	0.5225
*MMP9* mRNA	**−0.45 ± 0.17**	**0.0127**	0.14 ± 0.18	0.4319
*VEGF*-A mRNA	−0.15 ± 0.18	0.4084	−0.25 ± 0.17	0.1499

Separate general linear models were built with psychological improvement indices as outcomes, pre-rehabilitation measurand levels as continuous predictors, and the type of the psychometric test as a dichotomous covariate. The significant results are presented in bold.

**Table 5 brainsci-13-00846-t005:** Multivariate logistic regression performed to predict post-rehabilitation improvement in cognitive function based on pre-rehabilitation MMP-9 protein and mRNA levels. Age and sex are included as covariates. Hosmer–Lemeshow χ^2^(8) = 4.63, *p* = 0.7963, Nagelkerke R^2^ = 0.64, Cox-Snell R^2^ = 0.48.

Predictor	Odds Ratio		Wald Statistics	*p*-Value
	Point Estimate	95% CI		
MMP9 protein ^a^	1.34	1.04–1.73	5.2	0.0225
*MMP9* mRNA ^b^	0.25	0.08–0.76	6.0	0.0147
Age ^c^	0.90	0.78–1.04	2.0	0.1547
Sex ^d^	4.26	0.22–82.92	0.9	0.3391

^a^—odds for a growth in MMP9 protein concentration by 10 pg/mL, ^b^—odds for doubling the MMP9 mRNA expression, ^c^—odds for a growth in the age by a year, ^d^—odds for being male participants as compared to female participants.

**Table 6 brainsci-13-00846-t006:** Multivariate logistic regression predicting post-rehabilitation improvement in depressive symptoms based on pre-rehabilitation VEGF protein and mRNA levels. Age and sex are included as covariates. Hosmer–Lemeshow χ^2^(8) = 7.51, *p* = 0.4827, Nagelkerke R2 = 0.12, Cox-Snell R2 = 0.09.

Predictor	Odds Ratio		Wald Statistics	*p*-Value
	Point Estimate	95% CI		
VEGF protein ^a^	0.84	0.61–1.15	1.2	0.2693
*VEGF-A* mRNA ^b^	0.82	0.60–1.12	1.6	0.2109
Age ^c^	1.01	0.92–1.11	0.1	0.7922
Sex ^d^	1.17	0.24–5.76	<0.1	0.8438

^a^—odds for a growth in VEGF protein concentration by 10 pg/mL, ^b^—odds for doubling the VEGF-A mRNA expression, ^c^—odds for a growth in the age by a year, ^d^—odds for being male participants as compared to female participants.

## Data Availability

Not applicable.

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
