# Peer review of "Circulating Serum VEGF, IGF-1 and MMP-9 and Expression of Their Genes as Potential Prognostic Markers of Recovery in Post-Stroke Rehabilitation—A Prospective Observational Study"

_brainsci, 2023, doi:10.3390/brainsci13060846_

Round 1

Reviewer 1 Report

The study by WÅ‚odarczyk et al. entitled “Circulating Serum VEGF, IGF-1 and MMP-9 and Expression of their Genes as Potential Prognostic Markers of Recovery in Post-Stroke Rehabilitation- Prospective Observational Study” is straightforward and clear.

Overall, I found this article easy to read and full of analyses that convey its utility of MMP9 to be used as biomarker of cognitive improvement in post-stroke patients.  The only thing I would like to see is to improve English and formatting of the manuscript as there is evidence for the use of different fonts. 

Author Response

Dear Reviewer,

 Thank you for your revision, the answer is in the attachment.

Regards E.Miller et al

Reviewer 2 Report

The article is very interesting, well written and easy to understand. It presents a clear objective of the study, being solved with the obtained data. It presents a reduced number of patients, but it allows conclusions to be drawn.

The only problem with the article being the poor quality of the figures, namely 3. In addition, this should also have been better explained in the text, namely the program used to make it.

Author Response

(The authors gave the same response as above.)

Reviewer 3 Report

The authors analyzed in their work VEGF, IGF-1, and MMP-9 (genes and proteins) from the serum and the whole blood of patients 3-7 wk after the ischaemic stroke with moderate severity. These could be potential biomarkers in the prognosis of brain repair ability and thus could estimate the outcome of stroke. Moreover, they analyzed their associations with clinical scales including cognitive assessment and depression scales. 

My comments:

p-values should have the same decimals: please change this in the whole text and in the results. In addition, please add the p values as result in the abstract as well.

Figure3 needs higher resolution,

Author Response

(The authors gave the same response as above.)

Reviewer 4 Report

Dear Author,

Thanks for submitting your research manuscript entitled "Circulating serum VEGF, IGF-1 and MMP-9 and expression of their genes as potential prognostic markers of recovery in post-stroke rehabilitation - prospective observational study".

From my experience in the field, this manuscript is dealing with a very important Circulating serum VEGF, IGF-1 and MMP-9 and expression of their genes as potential prognostic markers of recovery in post-stroke rehabilitation and associated signaling pathway.

The author’s works conclude that the MMP9 (combined level of protein and mRNA expression) might be the useful biomarker of cognitive improvement in post stroke patient with 87% sensitivity and 71% specificity.

It is very interesting to confirm and identify biomarkers that can prognose brain repair ability and thus estimate the outcome of stroke by comparing the expression of VEGF, IGF-1 and MMP-9 proteins and their genes. I recommend the acceptance of this manuscript (with major comments) as it worth publishing and will add good information in the field of neuroscience.

Before giving my final comments and revising this manuscript, the author must address the following comments scientifically.
Reviewer  concerns:

Please find out the following comments:

·         The rationale and purpose behind selecting VEGF, IGF-1 and MMP-9 proteins and their genes and associated signaling pathway in correlation with recovery in post-stroke rehabilitation is unclear and need to reframe in introduction and discussion.

·         Updates old & outdated references.

plag percentage is very high. 

·         How was the sample size determined? Ideally, an a priori sample size calculation should be performed to determine the appropriate sample size.

·         Normality and variance homogeneity should be assessed across all groups of the same outcome variable, not individual experimental groups. Nonparametric tests must be performed if the data are not normally distributed or variance homogeneity is not met. Parametric data should be reported as mean +/- SD, while nonparametric data should be given/displayed as median and interquartile range. Longitudinal data should be analyzed using repeated measures tests.

-          Results need more clarification and significant justification. Differentiating between the outcome and the discussion sections is quite difficult.

-     To address the outcome of in-vivo measures/results separately, avoiding the disease condition and maintaining neurphysiological condition. How they correlate with the existing literature, it would be better if the author restructured to take a more critical approach for effective treatment to reduce neuronal complications.

-     In the discussion and the conclusion, the aims, rationale, and future perspectives are not evident clearly in relation to in-vitro and in-vivo experimentation.
-     The discussion is usually unorganized at the beginning to address and evaluate all the observations at the end. It makes the results easier to contextualize and more straightforward to comprehend.

- Furthermore, a minimal critical analysis should be provided, along with current study limitations and the future perspective as separate paragraphs.

-          Need to revise the conclusion scientifically. Not accepted in its current form.

-          A detailed revision shortening, ordering and following the commented ideas could improve this paper.

-          Several typewriting mistakes are present and need correction. This reviewer remains at the entire disposal for the next version.

Moderate editing of English language

Author Response

(The authors gave the same response as above.)

Round 2

Reviewer 4 Report

Dear Author, 

After careful revision, manuscript revised successfully. 

 Minor editing of English language required